# An Observational Study of MDR Hospital-Acquired Infections and Antibiotic Use during COVID-19 Pandemic: A Call for Antimicrobial Stewardship Programs

**DOI:** 10.3390/antibiotics11050695

**Published:** 2022-05-20

**Authors:** Nour Shbaklo, Silvia Corcione, Costanza Vicentini, Susanna Giordano, Denise Fiorentino, Gabriele Bianco, Francesco Cattel, Rossana Cavallo, Carla Maria Zotti, Francesco Giuseppe De Rosa

**Affiliations:** 1Department of Medical Sciences, Infectious Diseases, University of Turin, 10124 Turin, Italy; silvia.corcione@unito.it (S.C.); francescogiuseppe.derosa@unito.it (F.G.D.R.); 2Division of Geographic Medicine and Infectious Diseases, Tufts University School of Medicine, Boston, MA 02111, USA; 3Department of Public Health and Paediatrics, University of Turin, 10124 Turin, Italy; costanza.vicentini@unito.it (C.V.); carla.zotti@unito.it (C.M.Z.); 4S.C. Farmacia Ospedaliera, A.O.U. Città Della Salute e Della Scienza di Torino, 10126 Turin, Italy; sugiordano@cittadellasalute.to.it (S.G.); denisefiorentino21@gmail.com (D.F.); fcattel@gmail.com (F.C.); 5Microbiology and Virology Unit, Turin University, 10124 Turin, Italy; gabrielebnc87@gmail.com (G.B.); rossana.cavallo@unito.it (R.C.)

**Keywords:** hospital-acquired infections, antimicrobial stewardship, COVID-19, surveillance, resistance

## Abstract

The pandemic caused by the COVID-19 virus has required major adjustments to healthcare systems, especially to infection control and antimicrobial stewardship. The objective of this study was to describe the incidence of multidrug-resistant (MDR) hospital-acquired infections (HAIs) and antibiotic consumption during the three waves of COVID-19 and to compare it to the period before the outbreak at Molinette Hospital, located in the City of Health and Sciences, a 1200-bed teaching hospital with surgical, medical, and intensive care units. We demonstrated an increase in MDR infections: particularly in *K. pneumoniae* carbapenemase-producing *K. pneumoniae* (KPC-Kp), *A. baumannii*, and MRSA. Fluoroquinolone use showed a significant increasing trend in the pre-COVID period but saw a significant reduction in the COVID period. The use of fourth- and fifth-generation cephalosporins and piperacillin–tazobactam increased at the beginning of the COVID period. Our findings support the need for restoring stewardship and infection control practices, specifically source control, hygiene, and management of invasive devices. In addition, our data reveal the need for improved microbiological diagnosis to guide appropriate treatment and prompt infection control during pandemics. Despite the infection control practices in place during the COVID-19 pandemic, invasive procedures in critically ill patients and poor source control still increase the risk of HAIs caused by MDR organisms.

## 1. Introduction

The emergence of and subsequent pandemic caused by the COVID-19 virus have required major adjustments to healthcare systems and frameworks. As part of the response, infection control and antimicrobial stewardship programs (ASPs) have had to rapidly adapt in real time in the face of an evolving body of evidence [1]. 

According to a recent publication by Cassini et al. on attributable deaths and lost disability-adjusted life years caused by infections due to multidrug-resistant (MDR) bacteria in the European Union, Italy has a high estimated burden of antibiotic-resistant infections (201,584 cases, median) [2]. The recent COVID-19 pandemic has caused an increase in hospitalizations, creating a challenging situation in the healthcare system. Severe cases require prolonged hospitalization in intensive care units (ICU), where patients are at high risk for hospital-acquired infections (HAIs) due to the invasive procedures performed [3].

As reported in 2003 for the SARS-CoV epidemic, there was an increased rate of methicillin-resistant Staphylococcus aureus (MRSA) from 3.53% (3.53 cases per 100 admissions) during the pre-SARS period to 25.3% during the SARS period (*p* < 0.001), with an increased rate of ventilator-acquired pneumonia in ICUs, mostly (47.1%) caused by MRSA [4].

A recent review of the current medical literature showed that rates of bacterial or fungal co-infection reported in patients presenting with coronavirus infections appear to be low [5]. In nine studies reporting on bacterial co-infection in COVID-19 cases, bacterial/fungal co-infection was reported in 8% of cases; other reported incidences vary between 3.6 and 43% [6]. Many authors recognize the importance of superinfection, but definitive data are still lacking [7,8].

Antibiotic misuse coupled with a strained healthcare workforce and a reduced surveillance capacity for antibiotic-resistant organisms may lead to antimicrobial resistance as a lasting consequence of the COVID-19 pandemic [9,10]. Despite its viral nature, initial studies indicate that antibiotics are frequently prescribed to patients with COVID-19, largely due to suspected bacterial co-infections. Systematic reviews have estimated that more than 70% received antibacterial therapy, predominantly broad-spectrum and often empiric [11].

Understanding the patterns and predictors of antibiotic prescribing in COVID-19 can help to identify opportunities for interventions and target antibiotic stewardship strategies to improve the quality and safety of antibiotic use in the setting of the COVID-19 pandemic. Our objective was to compare antibiotic use and incidence of HAIs in the period covering the first through third waves of COVID-19 (1 February 2020–30 March 2021) with the pre-COVID period (1 August 2019–30 January 2020) to outline the changing epidemiology of nosocomial infections during the SARS-CoV2 pandemic.

## 2. Methods

### 2.1. Study Design and Objective

This is a retrospective and prospective observational epidemiological study at Molinette Hospital, located in the City of Health and Sciences, a 1200-bed teaching hospital in partnership with the University of Turin that provides various medical, surgical, and intensive care services. The study was approved by the Ethical Committee under protocol number 007831.

The pre-COVID period was defined as 6 months before the outbreak in Italy (1 August 2019–30 January 2020). The mentioned data were obtained retrospectively from hospital records. At the beginning of the COVID-19 pandemic, prospective data were collected as the pandemic evolved. The COVID period covered one year of the pandemic (1 Febuary 2020–30 March 2021) and included three waves.

### 2.2. Data Collection

Patients admitted to the hospital, including to surgical, medical, and intensive care units, from August 2019 to March 2021 were included if their stool sample was positive for *Clostridium difficile* (*C. difficile*) and/or their blood culture, respiratory culture (bronchoalveolar lavage or tracheal aspirate), or rectal swab was positive for MDR.

MDR organisms included *K. pneumoniae* carbapenemase-producing *K. pneumoniae* (KPC-Kp)*,* extended-spectrum beta-lactamase *Escherichia coli* (ESBL-*E. coli*), methicillin-resistant *Staphylococcus aureus* (MRSA), and carbapenem-resistant *P. aeruginosa and A. baumannii* (CR-PA; CR-AB). All positive blood and/or respiratory cultures for the previously mentioned organisms and *C. difficile* were reported. The absolute frequency and percentage were reported, as well as incidence ratios and rates expressed per 1000 patient-days (PD).

Monthly incidence of antibiotic use was calculated as defined daily dose (DDD). The DDD of carbapenems, vancomycin, fluoroquinolones (FQ), cephalosporins (CS), piperacillin–tazobactam, ceftazidime–avibactam (CAZ-AVI), and colistin were reported. Antibiotic use was obtained through pharmacy database records and normalized to defined daily doses (DDD) per 1000 PD.

### 2.3. Microbiological Detection Methods

Identification of bacterial isolates and antimicrobial susceptibility testing (AST) were performed on the Microscan Walkaway 96 plus system (Beckman Coulter, Brea, CA, USA) according to the EUCAST breakpoints. MRSA and ESBL phenotypes were detected according to the latest version of the “EUCAST guidelines for detection of resistance mechanisms and specific resistances of clinical and/or epidemiological importance”. Both were detected by broth microdilution tests integrated into the Microscan Walkaway panels. *Staphylococcus aureus* was considered MRSA by the positivity to the cefoxitin screening test (MIC > 4 mg/L) or if oxacillin MIC was >2 mg/L. ESBL production test was positive if a ≥8-fold reduction was observed in the MIC of cefotaxime, ceftazidime, or cefepime in combination with clavulanic acid (fixed concentration 4 mg/L) in comparison with the MIC of cephalosporins alone [12].

Carbapenemase production was detected by the phenotypic modified Hodge test and confirmed by immunochromatographic lateral flow tests (ICT) for rapid detection and typing of carbapenemases. Species identification was performed by MALDI-TOF MS.

### 2.4. Statistical Analysis

MDRO ratios and rates during the two time periods, prior to vs. during the COVID-19 pandemic, were compared using chi-squared tests and incidence rate ratios (IRRs), respectively. An interrupted time series analysis on trends in antimicrobial consumption (expressed as monthly DDD/1000 PD) was performed using an ARIMA model. The breakpoint was set on 1 February 2020 to investigate changes in antimicrobial usage prior to vs. during the COVID-19 pandemic. Estimates, standard errors, and *p*-value were reported. All analyses were performed using SPSS v. 27.0 (SPSS Inc., Armonk, NY, USA), and two-tailed statistical significance was set at <0.05.

## 3. Results

### 3.1. Microbiological Trends in Hospital

A significant increase in all MDR infections (*p*-value < 0.001) was observed comparing the COVID period to the previous period. KPC-KP isolates increased from 278 (14%) to 445 (23%) during COVID. ESBL-E. coli isolates increased from 173 (9%) to 225 (11.5%). *CR-AB* and *CR-PA* isolates increased from 26 (1.5%) to 199 (5%) and from 56 (3%) to 71 (4%), respectively. CDI rates increased from 92 (5%) to 130 (7%). The frequencies of HAP, BSI, and rectal swab isolates are reported in Table 1.

Adjusting per 1000 PD, we observed a significant increase in KPC-Kp during COVID from 3.41 to 4.46 isolates/1000 patient-days (IRR = 0.76) (Table 2). Similarly, *CR-AB* and MRSA isolates significantly increased during the COVID period from 0.32 to 2.00 isolates/1000 patient-days (IRR = 0.19) and 0.75 to 0.97 (IRR = 0.94), respectively. Rectal colonization by KPC-Kp and *CR-AB* was also higher during the COVID period, with IRRs of 0.97 and 0.2, respectively. On the other hand, the rates of ESBL-*E. coli*, *CR-PA*, and CDI were higher in the pre-COVID period, with IRRs of 1.14, 1.18, and 1.06, respectively. Further details are reported in Table 2.

### 3.2. Antibiotic Use

#### Overall Antibiotic Use

The use of first-generation cephalosporins and carbapenem saw a sharp decrease at the beginning of the COVID period, with a significant change in level (−539 and −564 DDD/1000 PD; *p* = 0.001 and *p* = 0.04, respectively). Although no significant trend was found, the pre-COVID downward trends were not maintained in the COVID period. FQ use showed a significant increasing trend in the pre-COVID period but saw a significant reduction in the COVID period (−204 DDD/1000 PD; *p* = 0.026). Figure 1 shows the time series analysis of the consumption of antibiotics, and Table 3 shows the estimates of DDD/1000 PD.

The use of fourth- and fifth-generation cephalosporins increased at the beginning of the COVID period, with a significant change in level (344 and 531 DDD/1000 PD; *p* = 0.02 and 0.002, respectively). Piperacillin–tazobactam use increased with a significant change in level (133 DDD/1000 PD; *p* = 0.053). No significant trends were identified in the pre-COVID or COVID periods for the use of fourth- and fifth-generation cephalosporins, piperacillin–tazobactam, and vancomycin. Colistin consumption decreased at the beginning of the COVID period (−87 DDD/1000 PD), but it was reversed to increase with a change in trend of 6.9 DDD/1000 PD during the COVID period. However, no significant trends were identified for colistin. Classifying per ward, no significant changes in trends were specifically observed in the medical, surgical, or ICU wards.

## 4. Discussion

The impact of the COVID-19 pandemic goes beyond the individual level, causing several public health issues, including antimicrobial resistance. One of the major concerns regarding hospitalized patients with COVID-19 are bacterial superinfections, especially in the ICUs and in patients with invasive devices, and outbreaks of MDR bacteria resulting from poor adherence to infection control practices. Earlier reports demonstrated HAIs in COVID-19 patients varying from 10–45% [13,14,15,16] but provided limited evidence about the microbial causality and the effect on outcomes. Some studies have described infections caused by MDR Gram-negative bacteria (*Enterobacterales*, *A*. *baumannii*, *P*. *aeruginosa*) in COVID-19 [17]; others have shown an increased number of co-infections due to the most common bacteria, such as *Mycoplasma pneumoniae*, *P. aeruginosa*, *H. influenzae*, and *Klebsiella* spp. [6].

This study described the incidence of MDR infections and changes in antibiotic use during the pandemic. Our data showed a reduction in MDR infections during the first wave of COVID-19, probably due to the increased use of personal protective equipment (PPE) and stronger adherence to infection control procedures. Moreover, the first COVID-19 wave might have benefitted from the active antimicrobial stewardship program that was ongoing at our institution, which had a positive influence on MDR trends in the last years before the pandemic [18].

Our data show that bacterial infections during the late periods of the pandemic increased; we demonstrated an increase in MDR infections in blood and respiratory cultures, particularly infections with with KPC-Kp, *CR-AB*, and MRSA, which is consistent with similar reports of patients with COVID-19 [19,20]. During COVID-19, there were important differences in the case mix: there was a greater proportion of clinically severe patients, while non-severe patients were encouraged to stay at home. Moreover, critically ill patients with COVID-19 required longer hospital stays and, consequently, posed a greater risk for acquiring nosocomial infections. This might have contributed to the HAI incidence ratio. In addition, our institute was a regional referral center for ExtraCorporeal Membrane Oxygenation (ECMO) in critically ill patients with COVID-19 acute respiratory distress syndrome (ARDS), and many patients were transferred to our facility from different hospitals. Those facts also might have affected the epidemiology and aided the MDR diffusion.

The high incidence of MDR could also be explained by the increased capacity of the hospital to admit all patients. This called for a great number of transferred healthcare staff who were probably inadequately familiar with the usual infection prevention practices. Another possible explanation is the high frequency of invasive procedures and a reduced focus on source control. Infection control strategies were strongly focused on preventing airborne viral pathogens, and less attention was probably paid to routine infection prevention activities (e.g., practices to prevent CLABSI and VAP, such as periodic evaluation of central lines, spontaneous breathing trials, and sedation interruption) [1,21].

Violations of the infection control protocols prompted by the increased workload and reduced availability of healthcare workers during the pandemic, combined with an institutional inability to sustain prevention and control measures, might have contributed to this finding.

In our study, we observed an increased consumption of antibiotics in the first wave of the COVID-19 pandemic, probably related to the earlier published data suggesting that COVID-19 be treated as community-acquired pneumoniae (CAP), using beta-lactams and azithromycin or FQ.

The lack of robust epidemiologic records in the initial stage of the pandemic, the relatively increased prevalence of bacterial co-infections (11–35%) in previous influenza outbreaks [22], and the involvement of co-infection with higher morbidity and mortality [23] perhaps affected physicians’ choice to start antibiotic therapy. This is further supported by the challenge of preventing a bacterial co-infection since the signs and laboratory and imaging results in COVID-19 overlap with pneumonia [24,25].

Later, the treatment conception was modified to eliminate the necessity of antibiotic coverage in COVID-19 patients except for critically ill patients and for patients with high clinical suspicion for bacterial infections [26,27]. This finding was also confirmed by our data, where antibiotic consumption did not significantly change during the second and third waves of the pandemic. A reduction in antibiotic use was perhaps not significantly achieved because our epidemiology of MDR was always endemic in our setting; therefore, we did not manifest major changes in antibiotic use.

During COVID-19, real-time collaboration between physicians, nurses, pharmacy, and infection control practitioners is demanded for antimicrobial stewardship. Infection control nurses incorporate elements of ASP into their infection control manuals and educational programs. ASP educational materials during the pandemic should include updates on local prescription practices, resistance patterns, and recommended approaches to treating common infections. Front-line nurses can also contribute to stewardship in numerous ways: obtaining blood cultures before antibiotic administration, the timely initiation of antibiotics in septic patients, adjustment and de-escalation of antimicrobials by communicating microbiology results to physicians, monitoring for adverse events related to antibiotics, reviewing orders for antibiotics, performing “time-outs” and assessing the need for ongoing antibiotic use, assessing for transition to oral antibiotics, and providing patient education. Pharmacists’ responsibilities include promoting optimal use through aiding in the appropriate selection, optimal dosing, rapid initiation, and proper monitoring and de-escalation of antimicrobial therapies, as well as the development of restricted antimicrobial use procedures, therapeutic interchange, de-escalation, treatment guidelines, and clinical care plans. Finally, the coordination between Microbiology, Infectious Diseases, and Preventive Medicine and Public Health should guide decisions regarding antibiotic use and the prevention of resistances [28,29].

One of the major strengths of our study include that it reported three waves of the COVID pandemic and evaluated invasive infections and colonization by MDR bacteria. Limitations of our study include the fact that it is a single-center study influenced by local epidemiology. Other limitations include the study’s descriptive nature as an epidemiological report without reporting clinical data, patient history, comorbidities, and therapies. We also did not outline the association between rectal colonization and developing invasive infection in this population, nor did we conduct an a priori sample size calculation.

In conclusion, our results describe the incidence of bacterial infections before and during the COVID-19 period and encourage the rapid re-establishment of antimicrobial stewardship to limit antimicrobial resistance. Despite the infection control practices in place during the COVID-19 pandemic, invasive procedures, prolonged hospital stays, and workload during the pandemic increased the risk of HAIs caused by MDR. Identifying changes in the resistance profile since the onset of the COVID-19 pandemic is an alarm for healthcare institutions and government agencies regarding the need to focus efforts on investigation and control, and to support and maintain a stewardship program and infection control practices.

## Figures and Tables

**Figure 1 antibiotics-11-00695-f001:**
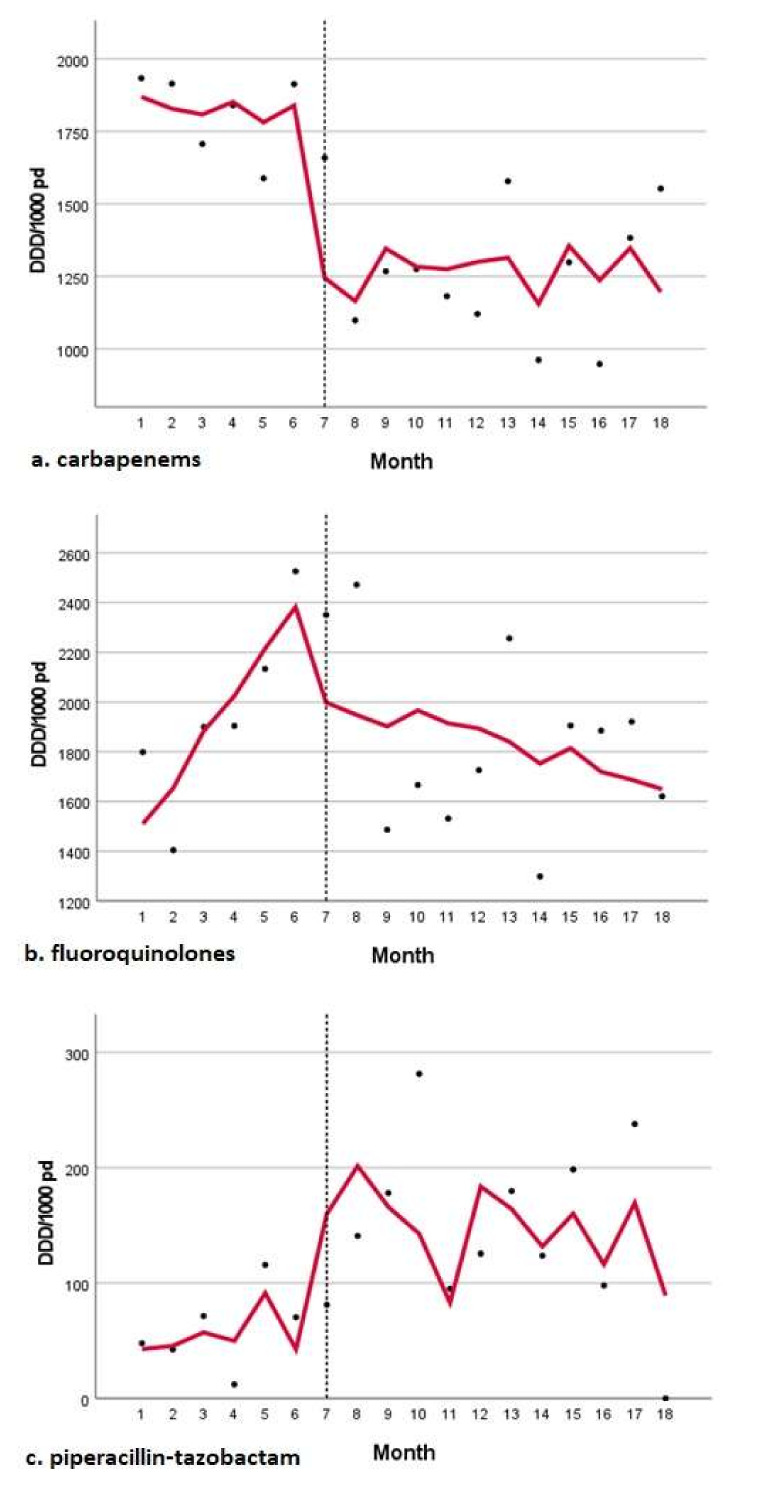
Time series analysis of antibiotic consumption. (**a**) carbapenem (**b**) fluoroquinolones (**c**) piperacillin-tazobactam.

**Table 1 antibiotics-11-00695-t001:** MDR isolates pre- and during COVID.

MDR	Total N (%)	Pre-COVID N (%)	COVID N (%)	*p*-Value
**KPC-Kp**	723 (36.9)	278 (14.2)	445 (22.7)	
Respiratory	111 (5.7)	33 (1.7)	78 (4)	<0.001
BSI	74 (3.8)	33 (1.7)	41 (2.1)	<0.001
RS	538 (27.4)	212 (10.8)	326 (16.6)	<0.001
**ESBL-*E. coli***	398 (20.3)	173 (8.8)	225 (11.5)	
Respiratory	45 (2.3)	17 (0.9)	28 (1.4)	<0.001
BSI	104 (5.3)	48 (2.4)	56 (2.9)	<0.001
RS	249 (12.7)	108 (5.5)	141 (7.2)	<0.001
**CR-AB**	225 (11.5)	26 (1.4)	199 (4.9)	
Respiratory	80 (4.1)	7 (0.4)	73 (3.7)	<0.001
BSI	27 (1.4)	4 (0.2)	23 (1.2)	<0.001
RS	118 (6)	15 (0.8)	103 (5.3)	<0.001
**CR-PA**	127 (6.5)	56 (2.9)	71 (3.7)	
Respiratory	104 (5.3)	47 (2.4)	57 (2.9)	<0.001
BSI	14 (0.7)	5 (0.3)	9 (0.5)	<0.001
RS	9 (0.5)	4 (0.2)	5 (0.3)	<0.001
**MRSA**	158 (8)	1 (3.1)	97 (5)	
Respiratory	69 (3.5)	24 (1.2)	45 (2.3)	<0.001
BSI	89 (4.5)	37 (1.9)	52 (2.7)	<0.001
**CDI**	222 (11.3)	92 (4.7)	130 (6.6)	<0.001

Abbreviations: BSI: blood-stream infections, RS: rectal swab, KPC-Kp: *K. pneumoniae* carbapenemase-producing *K. pneumoniae*, ESBL: extended-spectrum beta-lactamase, CR-AB: carbapenem-resistant *A. baumanii,* CR-PA: carbapenem-resistant *P. aeruginosa*, MRSA: methicillin-resistant *Staphylococcus aureus,* CDI: *Clostridium* difficile infection.

**Table 2 antibiotics-11-00695-t002:** Incidence rate ratios for adjusted isolates per 1000 patient-days.

MDR	Total/1000 PD	Pre-COVID/1000 PD	COVID/1000 PD	IRR	95% CI	*p*-Value
**KPC**	3.99	3.41	4.46	0.93	0.23–3.83	0.95
Respiratory	0.61	0.40	0.78	0.63	0.01–28.34	1
BSI	0.41	0.40	0.41	1.19	0.02–92.89	1
RS	2.97	2.60	3.27	0.97	0.19–4.95	1
**ESBL*-E. coli***	2.19	2.12	2.26	1.15	0.18–7.47	1
Respiratory	0.25	0.21	0.28	0.92	0.03–262.60	0.97
BSI	0.57	0.59	0.56	1.29	0.03–49.86	0.90
RS	1.37	1.32	1.41	1.14	0.10–12.28	1
** *A. baumanni* **	1.24	0.32	2.00	0.20	0.04–8.161	0.60
Respiratory	0.44	0.09	0.73	0.15	0.04–153.1	1
BSI	0.15	0.05	0.23	0.27	0.06–4209	0.77
RS	0.65	0.18	1.03	0.21	0.02–31.91	1
** *P. aeruginosa* **	0.70	0.69	0.71	1.18	0.04–32.62	0.90
Respiratory	0.57	0.58	0.57	1.24	0.03–48.10	0.90
BSI	0.08	0.06	0.09	0.81	0.02–2494	0.96
RS	0.05	0.05	0.05	1.22	0.05–295,300	0.97
**MRSA**	0.87	0.75	0.97	0.94	0.05–19.24	19
Respiratory	0.38	0.29	0.45	0.79	0.07–83.78	1
BSI	0.49	0.45	0.52	1.05	0.02–57.18	1
**CDI**	1.22	1.13	1.30	1.06	0.09–13.21	1

**Table 3 antibiotics-11-00695-t003:** Overall antibiotic trends in interrupted time series analysis in hospital.

Ward	Estimate	SE	*p*-Value
All wards			
*1st generation cephalosporins*			
Pre-COVID trend	−2.154	24.073	0.930
Change in level	−539.527	132.812	**0.001**
Change in trend	12.972	24.777	0.609
*2nd generation cephalosporins*			
Pre-COVID trend	−0.778	0.476	0.126
Change in level	−2.537	2.710	0.366
Change in trend	0.797	0.495	0.131
*3rd generation cephalosporins*			
Pre-COVID trend	−173.114	173.102	0.336
Change in level	−690.663	945.501	0.478
Change in trend	159.432	176.906	0.384
*4th generation cephalosporins*			
Pre-COVID trend	−19.712	21.995	0.386
Change in level	344.480	131.317	**0.021**
Change in trend	15.061	23.222	0.528
*5th generation cephalosporins*			
Pre-COVID trend	21.778	23.732	0.376
Change in level	531.828	134.128	**0.002**
Change in trend	−46.451	24.593	0.081
*Piperacillin–tazobactam*			
Pre-COVID trend	5.576	11.369	0.632
Change in level	133.944	62.834	**0.053**
Change in trend	−7.214	11.619	0.545
*Fluoroquinolones*			
Pre-COVID trend	173.061	77.478	**0.044**
Change in level	893.653	449.492	0.068
Change in trend	−204.120	81.122	**0.026**
*Vancomycin*			
Pre-COVID trend	−2.472	9.394	0.797
Change in level	−50.358	52.584	0.356
Change in trend	1.013	9.720	0.919
*Carbapenems*			
Pre-COVID trend	−19.560	44.567	0.668
Change in level	−564.805	247.041	**0.040**
Change in trend	15.009	45.425	0.746
*Caz-avi*			
Pre-COVID trend	2.472	9.394	0.797
Change in level	50.358	52.584	0.356
Change in trend	−1.013	9.720	0.919
*Colistin*			
Pre-COVID trend	3.023	11.170	0.791
Change in level	−87.292	67.925	0.221
Change in trend	6.923	11.923	0.571

## Data Availability

Due to patient confidentiality, raw data will be available upon request with a compelling reason.

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
