# Peer review of "An Observational Study of MDR Hospital-Acquired Infections and Antibiotic Use during COVID-19 Pandemic: A Call for Antimicrobial Stewardship Programs"

_antibiotics, 2022, doi:10.3390/antibiotics11050695_

Round 1
Reviewer 1 Report
This is a very important manuscript and Well conducted study. I am grateful for the opportunity to review this manuscript and only have minor comments for the authors:
- Add study design to the title
- Add study setting to abstract
- Define ICU in line 36
- You used HAI for two different definitions
- Line 75 use abbreviation ICU
- Add reference for methods in line 101
- Rename the name of Table 1
- Improve the quality of figure 1
- Align text cephalosporin In table 3 and add test used under the table
- Line 168 is this different font
- Add roles in AMR during COVID of other health care professionals (pharmacists, nurses) to the discussion section
- Add limitation section
Author Response
Dear Editor,
We would like to thank you for the opportunity to review our manuscript and we appreciate the valuable criticism. Please find the response to reviewers’ comments on our manuscript entitled "An observational study of MDR hospital-acquired infections and antibiotic use during COVID-19 pandemic: a call for antimicrobial stewardship programs" that was considered for publication in Antibiotics, and later encouraged for resubmission.
Comments of Reviewer 1
- Add study design to the title.
Dear reviewer, we highly value all your helpful comments. We have closely worked on addressing them.
Study design is now added to the title: "An observational study of MDR hospital-acquired infections and antibiotic use during COVID-19 pandemic: a call for antimicrobial stewardship programs”.
- Add study setting to abstract
The study setting: “Molinette Hospital, located in the City of Health and Sciences, a 1,200-bed teaching hospital in surgical, medical, and intensive care units” is added to the abstract.
- Define ICU in line 36
intensive care units (ICU) is adjusted.
- You used HAI for two different definitions
HAI abbreviation is now unified as hospital acquired infection throughout the manuscript.
- Line 75 use abbreviation ICU
ICU abbreviation is added.
- Add reference for methods in line 101
Reference to microbiological detection methods which is upon EUCAST guidelines is added.
- Rename the name of Table 1
Table 1 is renamed as: “MDR isolates pre- and during COVID”.
- Improve the quality of figure 1
I will confirm with the editing office about how to join the figures, which we also have as PDF with better quality, while maintaining the quality of pixels.
- Align text cephalosporin In table 3 and add test used under the table
Text of Table 3 was re-aligned. To my information, titles of tables are written up and titles of figures are written down. If I am wrong or this is against the format requested by the journal, I can confirm it with the editing office and change it accordingly.
- Line 168 is this different font
Apologies for that. It was fixed.
- Add roles in AMR during COVID of other health care professionals (pharmacists, nurses) to the discussion section
A paragraph about the role of front-line nurses, infection control nurses, and pharmacists is now added to the discussion section.
- Add limitation section
Limitations are added to the discussion.
Reviewer 2 Report
The authors aimed at comparing antibiotic use and incidence of HAI infections in the first to third wave of COVID-19 period compared to the pre-COVID period to outline the changing epidemiology of nosocomial infections during SARS-CoV2 pandemic. In general, the study is pertinent, easy to read and correct. Nevertheless, several changes are required:
1) Please read again the full manuscript in order to detect typos and mistakes. For example, double space in line 48, incidence OF HAI infections (line 59), K.pneumoniae carbapenemase producing-K.pneumoniae (line 79, the microorganism is repeated), etc. Wave of pandemia? (line 215): revise English throughout the manuscript.
2) All microorganisms should be in italics (e.g., lines 96-97 S. aureus). Revise thorughout the manuscritp.
3) How can a study be retrospective AND prospective? Both terms are incompatible, unless you did 2 studies and put the results together. Please explain and specify it in detail in the Methods.
4) Do not repeat the objetives in the methods (lines 69-70)
5) How can you compare 1 year of COVID period vs. 6 months of Pre-COVID? If you have double time, it is normal that you find more resistant microorganisms. Plase explain how did you consider this issue. How did you calculate incidence? What did you use as denominator?
6) Table 2 is not homogeneous. Please look that some numbers present one decimal, others present two decimals, some 95%CI are separated by commas or spaces... Revise deeply and correct.
7) Did you find any subgroup of patients with specially high incidence of MR bacteria or higher use of antibiotic stewardship? (for example, patients living in residences: https://doi.org/10.3390/antibiotics9060324 or patients from ICU https://www.frontiersin.org/articles/10.3389/fphar.2021.778386/full etc.)
8) Please include a large paragraph on limitations of the study in the Discussion.
Author Response
Dear Editor,
We would like to thank you for the opportunity to review our manuscript and we appreciate the valuable criticism. Please find the response to reviewers’ comments on our manuscript entitled "An observational study of MDR hospital-acquired infections and antibiotic use during COVID-19 pandemic: a call for antimicrobial stewardship programs" that was considered for publication in Antibiotics, and later encouraged for resubmission.
Comments of Reviewer 2
The authors aimed at comparing antibiotic use and incidence of HAI infections in the first to third wave of COVID-19 period compared to the pre-COVID period to outline the changing epidemiology of nosocomial infections during SARS-CoV2 pandemic. In general, the study is pertinent, easy to read and correct. Nevertheless, several changes are required:
1) Please read again the full manuscript in order to detect typos and mistakes. For example, double space in line 48, incidence OF HAI infections (line 59), K.pneumoniae carbapenemase producing-K.pneumoniae (line 79, the microorganism is repeated), etc. Wave of pandemia? (line 215): revise English throughout the manuscript.
Dear reviewer, we highly value all your helpful comments. We have closely worked on addressing them.
Double spacing, (KPC-Kp), waves of pandemia, are adjusted and the English was revised.
2) All microorganisms should be in italics (e.g., lines 96-97 S. aureus). Revise thorughout the manuscritp.
Italics for microorganisms is re-applied.
3) How can a study be retrospective AND prospective? Both terms are incompatible, unless you did 2 studies and put the results together. Please explain and specify it in detail in the Methods.
The pre-COVID period was the “retrospective” part; 6-months before the outbreak (01 Aug 2019- 30 Jan 2020). At the beginning of the COVID-19 pandemic, we started data collection “prospectively” as the pandemic evolved. This specification is added to the Methods.
4) Do not repeat the objetives in the methods (lines 69-70)
Repetition was deleted.
5) How can you compare 1 year of COVID period vs. 6 months of Pre-COVID? If you have double time, it is normal that you find more resistant microorganisms. Plase explain how did you consider this issue. How did you calculate incidence? What did you use as denominator?
The incidence ratios and rates were standardized per 1000 patient-days to avoid any bias in frequencies in different periods. The denominator was the sum of patient days in each period.
6) Table 2 is not homogeneous. Please look that some numbers present one decimal, others present two decimals, some 95%CI are separated by commas or spaces... Revise deeply and correct.
Table 2 was revised, and comments were addressed.
7) Did you find any subgroup of patients with specially high incidence of MR bacteria or higher use of antibiotic stewardship? (for example, patients living in residences: https://doi.org/10.3390/antibiotics9060324 or patients from ICU https://www.frontiersin.org/articles/10.3389/fphar.2021.778386/full etc.)
In fact, we have analysed data of antibiotic use in sub wards as well including: medicine, surgery and ICU. However, we didn’t observe significant difference. Therefore, we have stated the overall use only and reported in the last line of antibiotic use results: “Classifying per ward, no significant changes in trends were specifically observed in medicine, surgical or ICU wards.”
8) Please include a large paragraph on limitations of the study in the Discussion.
We elaborated the limitations that we had mentioned in the discussion:
“Limitations of our study include the fact that it is a single centre study influenced by local epidemiology. Other limitations include the descriptive nature as an epidemiological report without reporting clinical data, patient history, comorbidities, and therapies. We also did not outline the association between rectal colonization and developing invasive infection in this population.”
Round 2
Reviewer 2 Report
The authors made an effort to improve the manuscript and address my comments. However, several minor changes are still needed:
1) Please look at p-values in Table 3. They are sepparated by commas, except for colistin (sepparated by points). Use points throughout the Table for decimals (and commas for thousands).
2) The authors included a new paragraph (lines 226-241) on the role of nurses and pharmacists regarding antibiotic control. Note that a point after "de-escalation" should be deleted. Moreover, I have a suggestion. As several professions have now been highlighted, I believe that the role of Preventive Medicine and Public Health should also be recognised. For example, include a sentence similar to "Finally, the coordination between Microbiolgoy, Infectious Diseases and Preventive Medicine and Public Health should guide decisions regarding antibiotic use and prevention of resistances".
3) Include, if you agree, a limitation regarding sample size. I do not believe that this is a small sample size, but the absense of an "a priori" sample size calculation is a limitation.
Author Response
Dear Reviewer,
Thank you again for your comments. We highly value your extensive revision to improve the manuscript. Please accept our appologies for the minor issues that we have missed.
1) Please look at p-values in Table 3. They are sepparated by commas, except for colistin (sepparated by points). Use points throughout the Table for decimals (and commas for thousands).
All commas in Table 3 were replaced by periods since they are decimals.
2) The authors included a new paragraph (lines 226-241) on the role of nurses and pharmacists regarding antibiotic control. Note that a point after "de-escalation" should be deleted. Moreover, I have a suggestion. As several professions have now been highlighted, I believe that the role of Preventive Medicine and Public Health should also be recognised. For example, include a sentence similar to "Finally, the coordination between Microbiolgoy, Infectious Diseases and Preventive Medicine and Public Health should guide decisions regarding antibiotic use and prevention of resistances".
The point after de-escalation was deleted and replaced by a comma.
Thank you for this suggestion. This sentence was added at the end to highlight the role of Public Health. Indeed, this was what we applied since collegues from Public Health department were involved in this study and assisted in the authorship as well.
3) Include, if you agree, a limitation regarding sample size. I do not believe that this is a small sample size, but the absense of an "a priori" sample size calculation is a limitation.
We have added this to the Limitations also. We'll hopefully work on addressing sample size calculation in our future studies.